# How Organ-on-a-Chip Technology Can Assist in Studying the Role of the Glymphatic System in Neurodegenerative Diseases

**DOI:** 10.3390/ijms24032171

**Published:** 2023-01-21

**Authors:** Sarah Spitz, Eunkyung Ko, Peter Ertl, Roger D. Kamm

**Affiliations:** 1Faculty of Technical Chemistry, Vienna University of Technology, Getreidemarkt 9/163-164, 1060 Vienna, Austria; 2Department of Mechanical Engineering and Biological Engineering, Massachusetts Institute of Technology, Cambridge, MA 02139, USA

**Keywords:** AQP4, glymphatic system, microfluidics, organ-on-a-chip technology, neurodegeneration

## Abstract

The lack of a conventional lymphatic system that permeates throughout the entire human brain has encouraged the identification and study of alternative clearance routes within the cerebrum. In 2012, the concept of the glymphatic system, a perivascular network that fluidically connects the cerebrospinal fluid to the lymphatic vessels within the meninges via the interstitium, emerged. Although its exact mode of action has not yet been fully characterized, the key underlying processes that govern solute transport and waste clearance have been identified. This review briefly describes the perivascular glial-dependent clearance system and elucidates its fundamental role in neurodegenerative diseases. The current knowledge of the glymphatic system is based almost exclusively on animal-based measurements, but these face certain limitations inherent to in vivo experiments. Recent advances in organ-on-a-chip technology are discussed to demonstrate the technology’s ability to provide alternative human-based in vitro research models. Herein, the specific focus is on how current microfluidic-based in vitro models of the neurovascular system and neurodegenerative diseases might be employed to (i) gain a deeper understanding of the role and function of the glymphatic system and (ii) to identify new opportunities for pharmacological intervention.

## 1. The Glymphatic System 

In the peripheral tissues, the continuous exchange of interstitial fluid, an essential prerequisite to tissue homeostasis, is carried out by a coordinated interplay of the vascular and the lymphatic system. High arterial pressures generate a flux of fluid across the arterioles and capillaries of the vascular system into the surrounding tissues, generating a flow of interstitial fluid that either re-enters the circulation into the venules or flows to nearby lymphatic vessels, which ultimately return the fluid into the venous circulation. Thereby, nutrients are transported into the extracellular space while waste products are simultaneously cleared [1]. Interestingly, within the central nervous system, where imbalances in cerebral tissue homeostasis can have detrimental effects ranging from disturbances in synaptic cell signaling to the onset of neurodegenerative diseases, this conventional lymphatic network is restricted to the meninges, an arrangement of three membranes that envelopes the human cerebrum [2,3]. 

In vivo, the human brain is pervaded by a dense network of blood vessels comprising arteries, arterioles, capillaries, venules, and veins that, in addition to transporting nutrients and removing waste products, protect the nervous tissue from neurotoxic factors. To that end, the capillary network is equipped with a strictly regulated barrier, the blood–brain barrier (BBB), that, in contrast with the peripheral blood circulation, is composed of three distinct cell types: brain endothelial cells, pericytes, and astrocytes [4]. The high integrity of the BBB is conferred by a combination of tight endothelial cell–cell connections, ensheathing the pericytes, and the terminal end-feet of astrocytes positioned within the abluminal space of the blood vessels. To enable selective molecular transport across this tight barrier, endothelial cells are equipped with a range of transporter proteins, facilitating the transport of molecules into the brain parenchyma (influx) and vice versa (efflux) [4].

While the cerebrospinal fluid (CSF), a clear ultrafiltrate of plasma that surrounds the human brain, has long been thought to aid in the clearance of cerebral interstitial fluid, it was the identification of the meningeal lymphatics by MRI that made the presence of additional clearance routes increasingly evident [3]. The concept of an alternate clearance system was first proposed in 2012 by researchers from the Medical Center at the University of Rochester (Nedergaard Lab), who demonstrated the convective pathways of the CSF through the brain using two-photon imaging [5]. In detail, by intracisternal injection of fluorescently labeled CSF in mice, Iliff et al. monitored the subarachnoid influx of the CSF into the Virchow–Robin spaces along arterioles (1), followed by a directional influx of CSF into the parenchyma, where it mixed with the interstitial fluid (ISF) (2), before a subsequent efflux of the CSF/ISF mixture through the central veins thereafter (3) (see Figure 1). Furthermore, Iliff et al. showed that aquaporin-4 (AQP4) knockout mice displayed a 65% reduction in the CSF flux through the parenchyma, suggesting that the influx of water into the brain interstitium is controlled via the bidirectional water channel, predominantly expressed and located in the astrocytic endfeet (polarized expression) [5]. Based on these observations, the Nedergaard group proposed the term “glymphatic system“ in recognition of the glial-mediated water influx into the parenchyma that contributes to the compensatory lymphatic system within the brain [6]. While the proposed concept of an astrocyte-mediated clearance system initially produced some controversies, now, numerous studies have corroborated the presence of a perivascular waste-removal network [7] Intriguingly, recent studies have revealed that glymphatic activity is significantly enhanced during sleep, rendering sleep crucial in clearing neurotoxic waste and supporting the correlation between proteinopathies and sleep deprivation [7,8]. The increase in glymphatic flux can be linked to a decrease in the neurotransmitter norepinephrine, which has been shown to expand the extracellular space and thus result in a lowered hydraulic resistance [9]. In addition, a drastic age-dependent reduction in the glymphatic activity of 80-90% was observed in mice, providing a possible explanation for the increased vulnerability to developing neurodegenerative diseases with advancing age [10]. This decrease might be explained by the loss of polarization or depolarization of AQP4 (see Figure 2), which has been reported in astrocytes of aging mice and was linked to reduced cognitive performance in elder individuals (over 85 years old) [11,12]. Furthermore, both age-associated alterations in circadian rhythms and a decrease in CSF turnover rate might contribute to an abated clearance efficiency in the aging cerebrum [10,13,14].

Considering the critical role of the glymphatic system in maintaining cerebral homeostasis, fluctuations in this glial-mediated balance can consequently have detrimental effects. For this reason, over recent years, several studies have started to look into the potential role of the glymphatic system in neuropathologies, specifically within neurodegenerative proteinopathies, which comprise disorders characterized by abnormal aggregation of proteins as well as insufficient waste clearance. 

## 2. The Glymphatic System and Neuropathologies

A pathological inter-relationship between neurodegenerative proteinopathies and the glymphatic system was first reported by Iliff et al., who connected a marked decrease (70% for [^3^H]mannitol) in interstitial solute clearance with increased levels of amyloid β, the main constituent of plaques in Alzheimer’s disease, in an AQP4^-/-^ mouse model. Specifically, the authors reported a ~55% decrease in the rate of ^125^I-amyloid β _1–40_ clearance in AQP4^-/-^ mice compared with that in wild-type controls, indicating that a significant fraction of soluble amyloid β is cleared via perivascular routes as opposed to BBB-mediated local removal [5]. This hypothesis was recently supported by a study by Nauen and Troncoso, who demonstrated the presence of amyloid β for the first time in human lymph nodes. Compared with lymph nodes of the inguine, a greater abundance of amyloid β was reported in cervical lymph nodes, pointing toward the cervical lymph nodes as the primary entry point of amyloid β into the systemic lymphatics [15]. Furthermore, Peng et al. showed that chimeric human amyloid precursor protein (APP)/presenilin-1 (PS1) double-transgenic mice displayed reduced glymphatic clearance that preceded marked amyloid β deposition, rendering glymphatic influx a potential target for therapeutic intervention. Interestingly, the treatment of wild-type mice with amyloid β_40_ resulted in a significant reduction of CSF influx, pointing toward a negative AQP4-dependent reinforcement loop within Alzheimer’s disease [16]. The reported impaired glymphatic influx might be linked to depolarization of AQP4, which is observed in the post mortem tissues of Alzheimer’s patients. Interestingly, a comparative analysis of AQP4 in the CSF of Alzheimer’s disease patients and healthy controls revealed a 1.7× increase in AQP4, which positively correlated with tau levels [17]. Both SNTA1 and DAG1, which mediate the anchoring of AQP4 to the astrocytic endfeet, as well as MLC1, which encodes an astroglial membrane transporter linked to AQP4, positively correlated with tau levels, further relating the water channel AQP4 to tau pathology [18]. A study by Ishida et al. employing transgenic mice expressing P301S mutant tau showed that a deletion of AQP4 also markedly elevated tau levels within the CSF and significantly exacerbated p-tau deposition, aggravating the ensuing neuronal degeneration [19]. As hypothesized by Mogensen et al., these observations suggest an uncoupling of AQP4 expression and glymphatic influx, which might be explained by a concomitant depolarization of the water channel under pathological conditions [20]. Furthermore, as amyloid β oligomers and fibrils stimulate the release of proinflammatory cytokines in the microglia, inflammation constitutes an essential element in Alzheimer’s disease pathology [20,21,22]. Consequently, morphological changes in astrocytes mediated by astrogliosis might further enhance alterations within the glymphatic system [23]. Lastly, recent genetic studies have linked single-nucleotide polymorphisms (SNPs) within the AQP4 gene with protein depolarization, Aβ accumulation, disease stage progression, and cognitive decline, making SNPs potential predictors for the disease burden in an Alzheimer’s patients [24,25].

Alterations within the glymphatic system were also linked to Parkinson’s disease, the second most common neurodegenerative proteinopathy [26]. While α-synuclein, the protein affected within Parkinson’s disease, is traditionally assigned to the intracellular space of neurons, studies showed that α-synuclein can be excreted into the extracellular space and thereby might contribute to the interneuron transfer of protein aggregates [27]. The quantification of α-synuclein within post mortem brain tissue samples from Parkinson’s patients, displaying sleep disturbances, a common nonmotor symptom of the disease, revealed an increased synuclein burden within cerebral tissues compared with that of patients without impaired sleep patterns [28]. The potential inter-relationship between the glymphatic system and REM sleep, circadian rhythms, and clock gene dysfunction in Parkinson’s disease was recently summarized in a comprehensive review by Sundaram et al., and can be found elsewhere [29]. A potential involvement of the glymphatic system within Parkinson’s disease can also be derived from the multifaceted role of the affected neurotransmitter, dopamine, as the cholinergic monoamine transmitter was shown to both mediate ISF influx and directly modulate glial differentiation and AQP4 expression [30,31]. AQP4 deficiency, for example, was shown to enhance dopaminergic neurodegeneration and increase the susceptibility of TH-positive cells to the prodrug MPTP, a chemical inducer of PD in mice [32]. In addition, a potential interplay between the glymphatic system and Parkinson’s disease was supported by a study assessing α-synuclein levels in the CSF of patients and controls, revealing a 13% reduction in the unaggregated form of α-synuclein in Parkinson’s patients [33]. The hypothesis was further supported by a recent study conducted by Zou et al., who demonstrated perivascular α-synuclein aggregation and AQP4 depolarization in a mouse model of Parkinson’s disease (*A53T)*. In addition to glymphatic dysfunction, pathology-related phenotypes such as neurodegeneration and exacerbated α-synuclein aggregation were further enhanced upon the ligation of the cervical lymph nodes [34]. Furthermore, two recent studies have reported changes in the glymphatic system in two neurodegenerative disorders, Huntington’s and amyotrophic lateral sclerosis. Zamani et al. reported significantly disrupted glymphatic function in a transgenic mouse model emulating key pathological events implicated in amyotrophic lateral sclerosis; Wu et al. demonstrated reduced CSF–ISF exchange in a huntingtin (HTT)-expressing mouse model (BACHD) [35,36].

In to neurodegenerative diseases, alterations within the glymphatic system have also been reported for hydrocephalus, ischemic and hemorrhagic stroke, multiple microinfarctions, traumatic brain injury, cerebral amyloid angiopathy, as well as diabetes mellitus [37]. Hydrocephalus, for example, which is a pathological accumulation of CSF within the brain, has been associated with increased cerebrovascular pulsatility, decreased CSF influx, and impaired glymphatic clearance [38]. As a result, compounds modulating AQP4 function are currently under investigation in clinical trials [39]. Traumatic brain injury, a risk factor for neurodegenerative diseases, was shown to promote tau pathology in an AQP4-deficient mouse model [40]. In addition, reduced glymphatic influx was reported for both ischemic and hemorrhagic stroke, as well as multiple microinfarctions [37]. AQP4 was overexpressed at the site of infarction in ischemic stroke, and fibrin and fibrinogen deposits were shown to occlude perivascular spaces in hemorrhagic stroke, entrapment of CSF solutes was reported within small, dispersed ischemic lesions characteristic of multiple microinfarctions [37,41,42]. Impaired glymphatic transport was also reported for cerebral amyloid angiopathy, which, next to increased arterial stiffness, has been associated with a decreased arterial pulse and reduced perivascular spaces [43]. Lastly, an impaired glymphatic influx has also been observed in type II diabetes mellitus, a common metabolic disorder associated with cognitive impairment. Herein, the pathological phenotype is characterized by an increase in CSF influx, while interstitial solute clearance is reduced [44,45].

To summarize, over the last few years, substantial evidence has highlighted the critical role of the glymphatic system in maintaining cerebral homeostasis, and the detrimental effects of imbalances within this perivascular clearance system have been demonstrated. However, that most of our understanding of how the glymphatic system operates has been extracted from nonhuman in vivo models. While the development of new imaging techniques and post mortem studies corroborated initial hypotheses surrounding the glymphatic system, the limited access to the human cerebrovascular system necessitates alternative investigation strategies, preferably in the form of personalized in vitro models.

## 3. Organ-on-a-Chip Technology as a Tool to Model the Cerebrovascular System and Study Neurodegenerative Diseases

One research field with enormous potential for addressing the lack of suitable in vitro models is organ-on-a-chip (OoC) technology. OoC technology is defined as a transdisciplinary field encompassing elements of tissue engineering, cell biology, and, most importantly, microfluidics, a scientific discipline that focuses on the manipulation of minute amounts of fluids [46]. The term “OoC technology” was first coined in 2010 in a groundbreaking study by the research group of Donald Ingber, which demonstrated the vast potential of employing microfluidic engineering principles to develop microphysiological systems. Huh et al. presented a novel approach to mimic organ-level responses to bacteria and inflammatory cytokines in a human-cell-based model of the alveolar–capillary interface, incorporating both fluid flow and mechanical actuation [47]. Since then, an essential driving force behind the rapid development of OoC technology has been the strong need for reliable alternatives to conventional in vivo and in vitro models, specifically with regard to organ models with increasing complexity and realism and disease models. While in vivo animal models are restricted by high costs, ethical concerns, and interspecies differences, the earlier in vitro models, due to their substantial deviation from human physiology, were largely limited in their ability to mimic human (patho)physiologies involving multicellular interactions. OoC technology bridged this gap by having the ability to provide precise spatial and temporal control over cellular microenvironments to emulate (patho-)physiological tissue niches that account for (i) organotypic cellular arrangements and length scales, (ii) biological signaling among multiple cell types, and (iii) biophysical stimuli (e.g., compression, shear stress, etc.). Furthermore, by reducing reagent volumes and cell numbers and facilitating high-throughput compatibility, OoC technology remains more cost-effective [48,49,50]. Overall, the flexibility in design, material, and function, coupled with the capability of noninvasive monitoring (e.g., embedded microsensors), allows for the generation of custom-made platforms applicable to emulate and study any tissue of the human body. As such, it has already led to the successful recreation of various tissue analogs in vitro, such as the liver, gut, kidney, and lung [47,48,50,51,52,53].

## 4. Microfluidic Models of the Cerebrovascular System

Among the microfluidic models of the cerebrovascular system, one can distinguish between three different subcategories: (i) membrane-based models, (ii) single-lumen models, and (iii) self-assembled models (see Figure 3). Membrane-based models display the closest resemblance to conventional Transwell^®^ setups. Herein, the cells of the neurovascular unit are separated by an integrated porous membrane. While these primarily two-dimensional (2D) setups allow for intercellular communication and provide good controllability as well as comparability to Transwell^®^ (Corning, New York, United States) models, the cellular configuration and barrier properties within these systems differ from the in vivo physiological situation. Nonetheless, these models have provided valuable insight into the processes occurring within the neurovascular unit. Maoz et al., for example, employed a modular microfluidic setup to demonstrate the functional connectivity within the cellular constituents of the neurovascular unit [54,55]. Using this interconnected platform, the authors showed for the first time that the cellular constituents of the BBB and neurons are metabolically coupled. Pediaditakis et al. employed a membrane-based setup for studying the effects of dynamic fibrillar α-synuclein exposure on the neurovascular unit [55]. The authors were able to replicate key pathological phenotypes of Parkinson’s disease, including the formation of phosphorylated α-synuclein, mitochondrial impairment, neuroinflammation, as well as compromised barrier function. 

In addition to membrane-based models, microfluidic devices have been used to establish singular endothelial lumina in vitro. Herein, blood vessel mimics are generated by seeding endothelial cells into precoated channels or hollowed hydrogels. To better emulate the physiological microenvironment, the remaining cells of the cerebrovascular system are seeded next to or surrounding the endothelial lumen using various hydrogel matrices. Shin et al., for example, used a compartmentalized microfluidic device to enable endothelial cells lining a BBB vessel with a rectangular cross-section to communicate with neuronal cells that were genetically modified to overexpress amyloid β. Employing this setup, the authors were able to emulate pathological phenotypes in Alzheimer’s disease, including increased barrier permeability, reduced claudin 5 expression, and abluminal amyloid β deposition [56]. By coculturing endothelial cells, astrocytes, and neurons in a microfluidic device that enables intercellular communication through an array of parallelly arranged channels, Adriani et al. studied the inter-relationship between the three cell types of the neurovascular unit [57]. Seo et al. developed a model of high cellular heterogeneity by coculturing a cylindrical endothelial lumen with astrocytes, pericytes, neurons, oligodendrocytes, as well as neural stem cells [58]. The model, which was generated by forming a cylindrical hole in collagen hydrogels employing microneedles, displayed improved phenotypic characteristics as well as attenuated inflammatory responses after exposure to lipopolysaccharides. In a follow-up study, the authors introduced spheroids of two glioblastoma cell lines into the microfluidic platform to demonstrate the model’s applicability for studying BBB-associated chemosensitivity in glioblastoma [59]. 

Finally, a new microfluidic approach was developed for the BBB that utilizes the intrinsic self-assembling properties of cells within three-dimensional matrices. By coculturing primary/iPSC-derived brain endothelial cells, primary astrocytes, and primary pericytes in a precisely adjusted ratio within a fibrin hydrogel, Campisi et al. and Hajal et al. could mimic both the morphological and functional characteristics of the human BBB [52,60]. Using this method, the authors were able to emulate perfusable and in vivo like capillary networks with diameters ranging between 10 and 40 µm and permeability values (e.g., 4.2 × 10^−8^ cm/s for a 40 kDa dextran) comparable to those found in rat brains [60]. The model’s applicability was demonstrated in a study that elucidated the role of the CCL2–CCR2 astrocyte–cancer cell axis in cancer cell extravasation and brain metastasis [61]. For a more detailed overview of current models of the BBB, please refer to Hajal et al [62].

## 5. Microfluidic Models for Studying Neurodegenerative Diseases

In addition to models of the cerebrovascular system, OoC technology has been successfully employed to study pathological processes in neurodegenerative diseases (see Figure 4). Several studies, for example, focused on assessing the neurotoxic effects of amyloid β. Choi et al. demonstrated that oligomeric amyloid β confers higher toxicity than its fibrillar state under dynamic cultivation conditions. A correlation between concentration and toxicity was observed by establishing a concentration gradient of oligomeric amyloid β [63]. Park et al. exposed neurospheres to amyloid β under dynamic conditions, highlighting the inter-relationship between interstitial fluid flow and amyloid β toxicity [64]. Due to the ability of OoC technology to provide precise spatial control over cells, a particular focus of microfluidic studies has been directed toward analyzing axonal events. Song et al. proposed that the transport of amyloid β primarily occurs along the axonal membrane; Poon et al. showed that amyloid β oligomers could impair axonal transport, resulting in dysfunctional signaling of BDNF, an important neurotrophic factor [65,66]. Bianco et al. utilized spatial control to study the region-specific neurotoxic effects of amyloid β by separating fluidically connected cortical or hippocampal astrocytes from neurons [67]. Moreover, microfluidic platforms have also been successfully employed to emulate tauopathy. Deglise et al. presented an amyloid-β-mediated formation of pTau; Kunze et al. induced the formation of the post-translational modified protein via the addition of okadaic acid, a chemical that inhibits dephosphorylation within one group of interconnected neuronal populations [68,69]. Due to the improved optical accessibility in microfluidic devices, studies demonstrated both the internalization of aggregated pTau (via endocytosis) and its synaptic transfer, resulting in upregulated synaptic activity [70,71]. In addition, the migration behavior of neutrophils and microglia was assessed through the generation of amyloid β gradients [72,73]. Lastly, as mentioned in the previous section, Shin et al. functionally connected Alzheimer’s-disease-specific neurons to endothelial cells to analyze the effect of amyloid β on the integrity of a singular vascular lumen [56].

OoC technology also has been employed to investigate the pathological alterations associated with Parkinson’s disease. Microfluidic studies, for example, demonstrated that α-synuclein could be transported between individual neurons through axonal internalization. Furthermore, neuronal exposure to α-synuclein and its oligomeric species resulted in fragmentation and condensation of mitochondria, disruption of mitochondrial transport, as well as synaptic degradation [74,75,76]. This observation was further supported by those of Bolognin et al., who employed a microfluidic platform to study pathological phenotypes in three-dimensionally cultivated neuronal networks of patient-specific dopaminergic neurons carrying a G2019S mutation in the LRRK2 gene [77]. Van Laar et al. induced pathological phenotypes, which could be linked to upregulated mitochondrial DNA replication, via the introduction of rotenone, a broad-spectrum pesticide, insecticide, and environmental risk factor of Parkinson’s disease [78]. Wang et al. reported increased levels of α-synuclein oligomerization, seeding capability, and internalization upon inhibiting 14-3-3, an essential protein in α-synuclein trafficking, and thus pointed toward its potential involvement in α-synuclein propagation [79]. Moreover, microfluidic platforms have also been applied to investigate pathological processes in neurodegenerative diseases such as Huntington’s disease and amyotrophic lateral sclerosis. Zhao et al., for instance, reported impaired BDNF axonal transport in a corticostriatal network employing Huntington’s-specific murine neurons, which could be rescued after treatment with the chaperonin TriC [80]. Employing a comparable set-up, Virlogeux et al. confirmed Zhao et al.’s observations and further demonstrated a positive correlation between the maturation state of patient-specific neurons and impaired axonal transport. Next to increased mitochondrial transport along neuronal axons, decreased synaptic number and activity were observed in striatal neurons within patient-specific networks [81].

Lastly, in 2018, Osaki et al. developed an amyotrophic lateral sclerosis platform that interconnects patient-specific motor neuron spheroids (genetically engineered to express the light-sensitive channel rhodopsin-2) with engineered bundles of muscle fibers. Using this model, the authors were not only able to emulate disease-specific phenotypes, including reduced muscle contraction, increased apoptosis, and motor neuron degradation, but also to demonstrate drug-mediated rescue effects after the coadministration of bosutinib and rapamycin [82]. 

Parallel to the development of predominately 2D microfluidic models for investigating neurodegenerative diseases, significant progress has been made in employing OoC platforms for culturing human brain organoids. Hence, new opportunities have emerged to combine the advantages of organoid technology, namely its ability to generate complex, organotypic, and personalized tissue constructs, with the benefits of OoC models, ranging from spatial and temporal control over biophysical stimuli to noninvasive monitoring [83,84]. Cho et al., for example, demonstrated significant improvements in the functional maturation of human cerebral organoids embedded within a brain extracellular matrix enriched hydrogel using gravity-driven flow. In addition to the improved cellular viabilities and reduced necrotic cores, the dynamic culture markedly reduced organoid size variation from a coefficient of variance of 43.4 % to 17.4% [85]. Moreover, in a recent study by Spitz et al., OoC technology was employed for the first time to investigate pathological phenotypes in patient-specific organoids carrying a triple mutation of the α-synuclein gene, a common genetic alteration associated with familial Parkinson’s disease. Using a multisensor-integrated microfluidic platform, the authors could emulate and monitor Parkinson’s-disease-specific phenotypes, including dopaminergic neurodegeneration and Lewy body formation, which could be rescued in part following treatment with a repurposed compound [51].

## 6. Organ-on-a-Chip Technology as a Tool to Investigate the Role of the Glymphatic System in Neurodegenerative Diseases?

OoC technology has tremendous potential to provide new human-based in vitro models needed to further investigate the complex cellular interactions underlying function and pathologies associated with the glymphatic system. The broad array of existing microfluidic platforms allows us to study not only the arrangement of astrocytes along the capillary beds of the human BBB but also the effect of different flow patterns and speeds on cellular positioning as well as AQP4 polarization. Flow velocity can thus be varied to simulate, for example, the differences between the resting and active phases as well as age-related decreases in the interstitial fluid flow. The effects of pulsatile blood flow on the glymphatic system can be assessed by connecting vascular networks to specifically designed microfluidic pumps, capable of providing unidirectional flow for long periods of time [86]. While current self-assembled models of the capillary network offer a suitable starting point for the in vitro investigation of glial-dependent clearance, they lack penetrating arterioles, which is the first step along the glymphatic pathway and the main entry point of CSF into the cerebral interstitium. Recent advances in bioprinting, however, show promise for the study of penetrating arterioles and their surrounding Virchow–Robin spaces [87]. Szklanny et al., for example, engineered hierarchical vascular networks by combining a fenestrated singular vessel scaffold with a bioprinted hydrogel containing endothelial cells [88]. Furthermore, intricate structures of vascular networks could be mimicked by utilizing confocal images from tissues to guide the two-photon laser scanning lithography-based patterning of biomolecules [89]. A similar approach was utilized by Noor et al., who recreated vascularized cardiac tissues from patient biopsies, demonstrating the ability to generate patient-specific models [90]. Additive manufacturing methods have been employed to mimic individual blood vessels on a scale larger than capillary dimensions. Gao et al., for example, employed triple coaxial cell printing to spatially control the deposition of endothelial cells, smooth muscle cells, and fibroblasts [91]. While most of the published studies have been restricted to vessel diameters exceeding 600 µm, 10× larger than the penetrating arterioles in vivo, microvessels with effective diameters as small as 10 µm have already bene generated by exploiting laser-degeneration-based approaches [92,93]. Thus, while a combinatorial approach might be needed in order to recreate a microphysiological model of the full glymphatic system, merging bioprinting with OoC technology provides a robust set of tools for the study of the cerebral clearance system. 

Control over spatial cellular arrangements and microenvironments conferred by OoC technology, coupled with the ability of brain organoids to emulate complex (pathological) phenotypes in vitro, constitutes an exemplary basis for investigating the role of the glymphatic system in the context of neurodegenerative diseases. Today, a broad selection of OoC and organoid-based models emulating neurodegenerative processes has been developed, ranging from Alzheimer’s disease models to organoids emulating Parkinson’s-disease-associated phenotypes, to models mimicking Huntington’s disease and amyotrophic lateral sclerosis. By employing patient-specific cells associated with early onset familial Alzheimer’s disease, Raja et al., for example, were able to mimic the critical hallmarks of this neurodegenerative disease, including aggregation of amyloid β, hyperphosphorylated tau, and abnormal endosomes [94]. By generating organoids from patient-specific cells carrying a mutation in presenilin-1, Yan et al. demonstrated the secretion of amyloid β_42_, a key peptide associated with Alzheimer’s disease [95]. Significant progress has also been made in the development of patient-specific organoid or OoC models capable of emulating the essential processes of Parkinson’s disease. Advancements in the modeling of the neurodegenerative proteinopathy in vitro have encompassed the replication of key pathological phenotypes such as dopaminergic neurodegeneration, aggregation of α-synuclein, Lewy-body formation, mitochondrial impairment, as well as reduced astrocytic activity [51,96,97,98]. While the number of models emulating Huntington’s disease and amyotrophic lateral sclerosis is still limited, the first successful protocols have already been reported. By deriving cortical organoids from the iPSCs of a Huntington’s disease patient, for example, Conforti et al. demonstrated aberrant neurodevelopment, as characterized by disrupted cytoarchitecture, abnormal rosette formation, failure of neural-ectodermal acquisition, and an immature ventricular/subventricular zone [99]. Szebényi et al. were able to mimic the early molecular pathology of amyotrophic lateral sclerosis overlapping with frontotemporal dementia by utilizing patient-specific cells harboring the C9ORF72 hexanucleotide repeat expansion mutation [100]. Additionally, Osaki et al. were able to show differences in the innervation, force generation, and cell death in engineered muscle bundles activated by motor neurons derived from a patient expressing a TDP43 mutation [101]. 

Overall, organoid technology provides a broad range of models that can readily be interconnected to cellular replicas of the cerebrovascular system employing microfluidic engineering principles. Thus, it provides a promising basis for investigating specific aspects of the complex interplay between the glymphatic system and neurodegenerative diseases, including, for example, pathology-mediated depolarization of AQP4, changes in AQP4 expression, as well as AQP4-mediated clearance of essential pathological proteins. Moreover, by integrating organoids into perfusable microfluidic platforms, the effects of various interstitial flow patterns (e.g., resting vs. awake states) on clearance and toxicological impact might be assessed without the need for any preceding coupling to models of the cerebrovascular system. Furthermore, recent advances in modeling the lymphatic system also have made conceivable the investigation of the interaction between the glymphatic system or glial-mediated interstitial fluid flow patterns, and the meningeal lymphatics [101,102].

## 7. Conclusions

The increasing body of literature around the glymphatic system has clearly demonstrated its essential role in maintaining cerebral homeostasis as well as its potential involvement in the onset and exacerbation of a variety of neuropathologies. While in vivo imaging strategies and post mortem studies on humans have corroborated the proposed concept of a glial-mediated clearance system, most of the current information has been derived from animal models. Considering interspecies differences and the fact that specific pathologies such as Parkinson’s disease are restricted to humans, alternative in vitro models that provide a better level of control and that work on a patient-specific basis are urgently needed. OoC technology, with its ability to offer precise spatiotemporal control over cellular microenvironments, including biophysical stimuli, provides an optimal basis for addressing this limitation. Recent advances in the field over the last few years have enabled, among others, the mimicking of the cerebrovascular system in vitro, the study of pathological processes in neurodegenerative diseases, as well as the emulation of neuronal tissues and networks. These provide a solid foundation for the investigation of the glymphatic system outside conventional in vivo settings. Soden et al. very recently demonstrated the potential of this transdisciplinary approach in their study [103]. Therein, the authors presented the first microfluidic coculture model to study the effect of lipopolysaccharides, amyloid-β oligomers, and AQP4 inhibition on glymphatic drainage. The study showed not only that fluidic drainage was AQP4-dependent but also that the inhibition of the latter in vitro resulted in impaired fluid clearance. As such, the study provides a perfect practical example of how microfluidics can be employed to investigate the glymphatic system ex vivo. While the in vitro generation of biomimetic replications of the glymphatic system still requires further transdisciplinary efforts, a variety of existing models can readily be employed to investigate specific processes underlying the glial-mediated clearance system. Moreover, by integrating cells from other species (e.g., rodents or nonhuman primates), the applicability of organ-on-a-chip technology in the context of the glymphatic system may be tested and validated with the results of past in vivo studies. Concurrently, organ-on-a-chip-derived data should be compared with clinical data to further assess its level of clinical mimicry, an essential prerequisite in ensuring the technology’s translatability. We hope that this review will accelerate this process and provide potential reference points for future studies focusing on unraveling the key role of the glymphatic system in cerebral health and disease. 

## Figures and Tables

**Figure 1 ijms-24-02171-f001:**
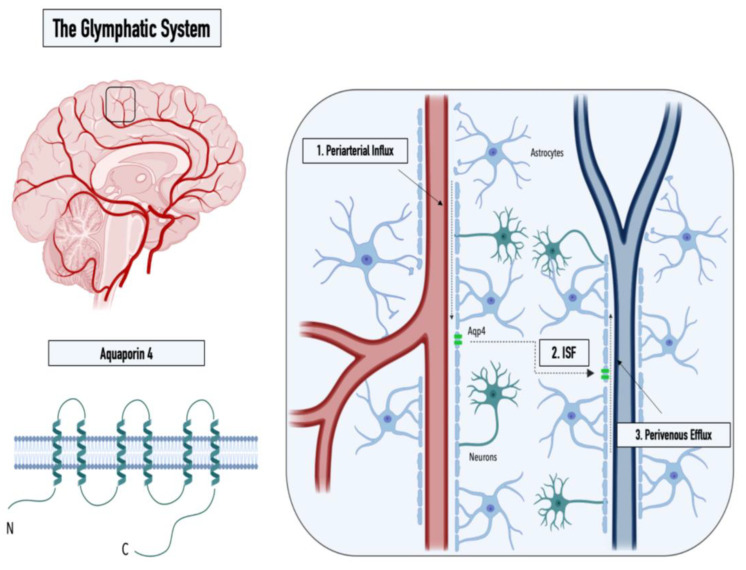
Graphical illustration of the structure of the water channel aquaporin-4 (**left** panel) and the three steps of the glymphatic pathway: (**1**) CSF influx into the Virchow–Robin spaces along arterioles, (**2**) influx of CSF into the parenchyma, and (**3**) perivenous efflux of the CSF/ISF mixture (**right** panel).

**Figure 2 ijms-24-02171-f002:**
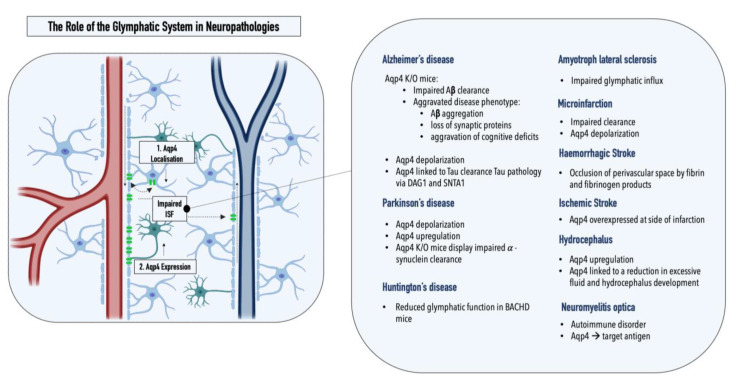
Graphical illustration of the two key pathological alterations in the glymphatic system observed in neuropathologies: (**1**) AQP4 depolarization and (**2**) altered AQP4 expression. Overview of pathological changes in the glymphatic system observed in various neuropathologies.

**Figure 3 ijms-24-02171-f003:**
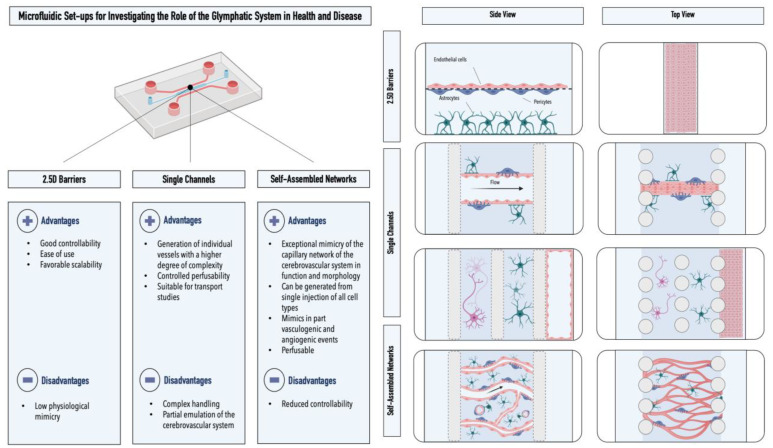
Advantages and disadvantages of microfluidic models of the cerebrovascular system. Graphical illustrations represent the three different microfluidic setups utilized for the emulation of the cerebrovascular system in vitro: 2.5D barrier set-ups, single-channel devices, and self-assembled networks. Astrocytes are depicted in green, endothelial cells in pink, and pericytes in lilac.

**Figure 4 ijms-24-02171-f004:**
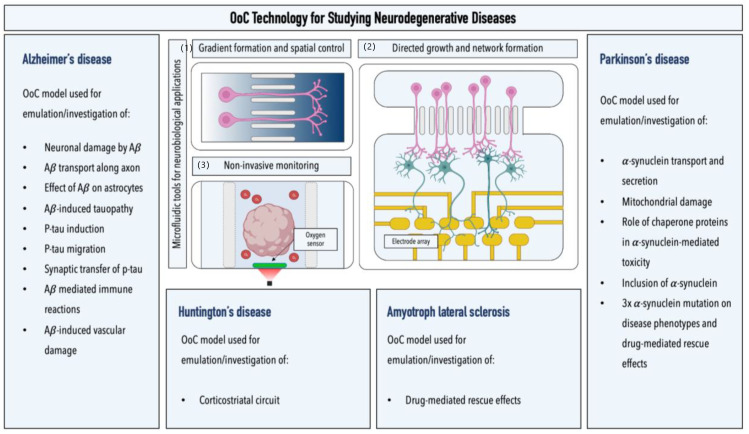
Overview of microfluidic models for studying neurodegenerative diseases. Graphical illustrations highlight the advantages of microfluidic setups for studying neurodegenerative diseases: (**1**) generation of biochemical gradients (image illustrates two interconnected microfluidic channels with varying concentrations); (**2**) noninvasive monitoring of cells of the neurovascular unit by the integration of electrical, optical, and electrochemical sensors (image illustrates a brain organoid-on-a-chip platform equipped with integrated luminescent oxygen sensor spots for recording cellular respiration); and (**3**) directed growth and network formation (image illustrates two interconnected microfluidic chambers that allow for the directed growth and functional connection of two spatially separated neuronal populations; an integrated multielectrode array allows for the electrophysiological recording of neuronal activity in the downstream chamber of the device).

## Data Availability

Not applicable.

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
