# Peer review of "How Organ-on-a-Chip Technology Can Assist in Studying the Role of the Glymphatic System in Neurodegenerative Diseases"

_ijms, 2023, doi:10.3390/ijms24032171_

Round 1

Reviewer 1 Report

The review by Spitz and co-worker is very interesting since it concerns two innovative issue: the role of glymphatic system in neurodegeneration and its modeling by using alternative methods to substitute animal models in accordance to the 3R principle. The review is well-written and only a minor revision is necessary:

1. Abbreviation should be added in the figure legends; in particular, in figure 1 and 2 IFF is not explained

2. Interstitial Fluid abbreviation should be aligned since it was used in different forms (IF, IFF, ISF);      ISF appears the most appropriate

3. on line 108 , Nauen and Troncoso, “et al” should be deleted

4. On line 272 Campisi et al and Hajal et al, “and” is not italic

5. Correct some words highlighted in revision mode on line 292 (effects) and 376 (pumps)

6. On line 381 change “Robin-Virchow” in “Virchow-Robin” as used in the rest of the review

7. Check all references because informations are missing (journal name, issue, volume and page numbers where necessary)

Author Response

We thank the reviewer for the feedback. We have corrected and adapted the manuscript accordingly. The revised parts have been highlighted in the new manuscript. 

Reviewer 2 Report

The authors provide a review on a timely and important field of research. Indeed, the role of glymphatic system in CNS health and disease is currently highly acknowledged and new technologies, including ‘organ-on-a-chip’ emerged within the last decade as a valuable tool to study it. The review provides a fair overview of the literature data in an analytical approach discussing both advantages and disadvantages of each particular technological tool. The manuscript is well written and easy to follow and would be certainly of interest for a broad readership.

I have only a few minor suggestions:

1. It may be interesting to provide a brief perspective on how the data obtained with these new ‘organ-on-a-chip’ technologies could be validated in more integrated models in vivo (using non-human primates? other?) before translating the knowledge to human clinics.

2. Figure are nice and illustrate well the discussed key findings. However, the legends should be more detailed by, for example, naming the cell types in the Fig 3. Similarly, what exactly represent the yellow objects in ‘Directed growth and network function’ part of the Fig 4? Are they depicting the electronic device purposed to mimic neuronal network / recording device / other?

3. AQP4 depolarization is largely discussed along the review but it is not explicitly defined to what it refers and what are precisely its functional consequences. In my opinion, including this information would be helpful for less specialized readers.

Author Response

 MC 1) It may be interesting to provide a brief perspective on how the data obtained with these new ‘organ-on-a-chip’ technologies could be validated in more integrated models in vivo (using non-human primates? other?) before translating the knowledge to human clinics.

R1: We agree with the reviewer and have included the following brief perspective regarding the potential validation of organ-on-a-chip platforms in the context of the glymphatic system in the revised manuscript:

 ..” Moreover, by integrating cells from other species (e.g., rodents or non-human primates) the applicability of organ-on-a-chip technology in the context of the glymphatic system may be tested and validated with past in vivo studies. Concurrently, organ-on-a-chip-derived data should be compared to clinical data to further assess its level of clinical mimicry, an essential prerequisite in ensuring the technology’s translatability.”..

MC 2). Figure are nice and illustrate well the discussed key findings. However, the legends should be more detailed by, for example, naming the cell types in the Fig 3. Similarly, what exactly represent the yellow objects in ‘Directed growth and network function’ part of the Fig 4? Are they depicting the electronic device purposed to mimic neuronal network / recording device / other?

R2: We have adapted the figures and captions accordingly.

MC 3). AQP4 depolarization is largely discussed along the review but it is not explicitly defined to what it refers and what are precisely its functional consequences. In my opinion, including this information would be helpful for less specialized readers.

R3: We agree with the reviewer. The term polarization and depolarization are explained in more detail in the revised manuscript:

..“Furthermore, Iliff et al. showed that aquaporin-4 (AQP4) knockout mice displayed a 65% reduction in the CSF flux through the parenchyma, suggesting that the influx of water into the brain interstitium is controlled via the bidirectional water channel, predominantly expressed and located in astrocytic endfeet (polarized expression).5 “..

..”This decrease might be explained by the loss of polarization or depolarization of AQP4, respectively (see Figure 2), which has been reported in astrocytes of aging mice and was linked to reduced cognitive performance in elder individuals (over 85 years old).12,13”..
